# Deciphering the pathogenicity of three *NKX2-1* variants in ultra-severe forms of childhood interstitial lung disease

Yohan David Soreze[1,2]*, Tifenn Desroziers-Louedec[1], Aurore Carré[3], Farah Diab[1], Aphrodite Daskalopoulou[1], Julie Starck[2], Valérie Nau[4], Marie Legendre[1,4], Sonia-Athina Karabina[1], Véronique Houdouin[5], Aurore Coulomb-L'herminé[6], Camille Louvrier[1,4☯], Nadia Nathan[1,7☯]

**1** INSERM, UMR_S933 Childhood genetic diseases, Armand Trousseau Hospital, Paris, France, **2** Assistance Publique Hôpitaux de Paris, Intensive care unit, Armand Trousseau Hospital, Paris, France, **3** Cochin Institute, INSERM U1016 and Imagine Institute Affiliate, Paris, France, **4** Assistance Publique Hôpitaux de Paris, Medical Genetics Department, Armand Trousseau Hospital, Paris, France, **5** Assistance Publique Hôpitaux de Paris, Pediatric Pulmonology department, Robert Debré Hospital, Paris, France, **6** Assistance Publique Hôpitaux de Paris, Pathology Department, Sorbonne University APHP, Armand Trousseau Hospital, Paris, France, **7** Assistance Publique Hôpitaux de Paris, Pediatric Pulmonology department, Reference center for rare lung diseases (RespiRare), Armand Trousseau Hospital, Paris, France

☯ These authors are contributed equally on this work.
* yohan.soreze@aphp.fr

## Abstract

### Introduction

The transcription factor NK2 homeobox1 (NKX2−1), associated with brain lung thyroid syndrome, regulates the transcription of surfactant proteins, thyroglobulin (TG) and thyroid peroxidase (TPO). This study explored the pathogenicity of three *NKX2−1* variants (p.(Tyr214Cys), p.(Arg165Trp) and p.(Gly147Ala)) that were identified in three infants with lethal forms of childhood interstitial lung disease.

### Methods

HEK293T cells were co-transfected with expression plasmids of NKX2−1 (wild-type (WT) and variants) and PAX8, along with reporter plasmids containing the promoters of *SFTPB, SFTPC, TG* and *TPO*). Protein expression was analyzed by western blotting and immunofluorescence. Luciferase assays were performed to evaluate the activation of different promoters. Surfactant protein and NKX2−1 expression were also assessed on patient lung biopsies using immunohistochemistry.

### Results

All three mutant proteins exhibited nuclear localization. Protein expression was altered in the p.(Tyr214Cys) and p.(Arg165Trp) variants located in NKX2.1 homeodomain. The p.(Tyr214Cys) variant failed to transactivate the tested promoters and was associated

**Data availability statement:** The minimal anonymized dataset necessary to replicate my study findings are in the supporting information and can be found with the following DOI: 10.5281/zenodo.17739209.

**Funding:** The author(s) received no specific funding for this work.

**Competing interests:** NN: Grants or contracts from any entity 2024: CORTICONEHI: Clinical trial: Efficacy of methylprednisolone pulses in neuroendocrine cells hyperplasia of infancy: an early phase study. 2023: Million Dollar Bike Ride project for Neuroendocrine Cell Hyperplasia of Infancy: Genetic basis of neuroendocrine cell hyperplasia of infancy 2022: Chancellerie des Universités: Legs Poix, Molecular and phenotypic characterization of interstitial lung disease n°2022000594 2022: RespiFIL grant for the development of an e-learning module for CT-scan in childhood interstitial lung diseases (15 000 €) 2022: RespiFIL grant for the development of an online platform for the collection of quality of life and transition questionnaires in rare lung disease (15 000 €) Payment or honoraria for lectures, presentations, speakers bureaus, manuscript writing or educational events 2022: La lettre du Pneumologue Support for attending meetings and/or travel: 2023: ERS Travel grant Leadership or fiduciary role in other board, society, committee or advocacy group, paid or unpaid 2021-2025 Head of the ERS Clinical research collaboration for childhood ILD (CRCchILDEU) 2017-2023 Treasurer of the Société française de pédiatrie (SFP) 2023-Treasurer and Scientific committee of the Société de Pneumologie Pédiatrique et d'Allergologie (SP2A) 2023- Scientific and Scientific committee of the Société de Pneumologie de Langue Française (SPLF) AC: All support for the present manuscript Supply of vectors This does not alter our adherence to PLOS ONE policies on sharing data and materials.

with a lack of pro-SP-C and SP-C expression in lung biopsy whereas the p.(Arg165Trp) variant induced both gain- or loss-of-function effects on the tested promoters. Finally, the p.(Gly147Ala) variant transactivated all the promoters tested, as for the WT. Conclusion: Our results demonstrated the pathogenicity of two variants, p.(Tyr214Cys) and p.(Arg165Trp), located within the homeodomain of NKX2−1. Conversely, the p.(Gly-147Ala) variant showed no pathogenic effects. To date, the p.(Tyr214Cys) variant is associated with the most severe respiratory phenotype reported for NKX2–1-related disorders. Further studies are needed to understand the specific mechanisms underlying the pathogenicity of NKX2.1 variants located in the homeodomain.

## Introduction

The transcription factor (TF) NK2 homeobox 1 (NKX2−1, also called thyroid transcription factor TTF-1) is encoded by the *NKX2−1* gene and partially controls the synthesis of surfactant proteins [1,2]. NKX2−1 was initially identified as a TF in the thyroid gland, before Lazzaro et al. demonstrated its expression in the lungs and brain as well [3]. Two isoforms of *NKX2−1* exist: the short isoform (NM_003317.3), which predominates in the lungs and thyroid, and the long isoform (NM_00107966.2), expressed in all three tissues [1,4]. In the lungs, NKX2−1 regulates architectural patterns and the transcription of genes encoding the four surfactant proteins and the ATP-binding cassette sub-family A member 3 (ABCA3) transporter [2,5–7]. In the thyroid, NKX2−1 activates transcription of thyroglobulin (*TG*) and thyroid peroxidase (*TPO*) [8]. The binding site of NKX2−1 on the *TG* and *TPO* promoters overlaps with that of paired box gene 8 (PAX8), another TF essential for thyroid development and function [9,10]. In the brain, *NKX2−1* is expressed in hypothalamic neurons and participates in interneuron specification and migration during forebrain development [8].

Heterozygous pathogenic variants of *NKX2−1* can be associated with brain-lung-thyroid syndrome. The complete syndrome comprises neurological symptoms, typically neonatal hypotonia and benign chorea, peripheral congenital hypothyroidism (CH), and respiratory symptoms ranging from severe neonatal respiratory distress to interstitial lung disease (ILD). This phenotype is consistent with *NKX2−1* knockout mice that can exhibit an incomplete lung parenchyma, an incomplete thyroid gland, and defects in neurological structures [1, 11]. In humans, some patients exhibit only partial or delayed forms of the disease, involving only one or two organs [4,8,12].

In a previous study, we reported a case of a full-term female neonate with acinar dysplasia and a likely pathogenic *NKX2−1* variant [13]. Based on this report, we aimed to analyze the functional consequences of this variant, along with two other *NKX2−1* variants, identified in infants with ultra-severe lung disease.

## Results

### Patients and molecular analysis

Patient 1, the most severe case, has been extensively reported [13]. In summary, her parents were originally from Togo and she presented with severe neonatal respiratory

distress (Fig 1A and S1 Fig) and pulmonary hypertension (PHT), and was further diagnosed as acinar dysplasia. The patient also presented with neonatal hypotonia and peripheral CH with a normal-sized and normally located thyroid gland (S2 Fig). The patient exhibited hypotonia before being sedated for mechanical ventilation and extra corporeal membrane oxygenation (ECMO). Patient features are displayed in Table 1.

Immunostaining of the lung was performed to assess NKX2−1 expression on patient lung biopsies, revealing weak and heterogeneous expression of mature SP-B and its precursor pro-SP-B, absent expression of SP-C and pro-SP-C, and normal NKX2−1 expression within hyperplastic alveolar epithelial cells (Fig 1D). Targeted Sanger sequencing (S3 Fig) revealed a heterozygous missense *NKX2−1* variant c.641A>G p.(Tyr214Cys), which was absent from the population database GnomAD v4.1.0, and predicted to be damaging (Alphamissense 1, REVEL 0.94). This variant affects a conserved amino acid within the homeodomain of NKX2−1 and was classified as likely pathogenic according to the American College of Medical Genetics and Genomics criteria [14] (Fig 2). The variant was not detected in the mother, and paternal DNA was unavailable for analysis. The patient died at 5 weeks of age, following the discontinuation of ECMO support.

Patient 2 was a full-term male infant of Sub-Saharan African descent with no significant medical history until 2.5 years of age, except for failure to thrive. He was unknown from our center until he first presented with a severe Respiratory Syncytial Virus (RSV) infection leading to respiratory failure and hospitalization in the intensive care unit with noninvasive ventilation. Due to atypical progression and clinical signs of chronic respiratory insufficiency, such as digital clubbing

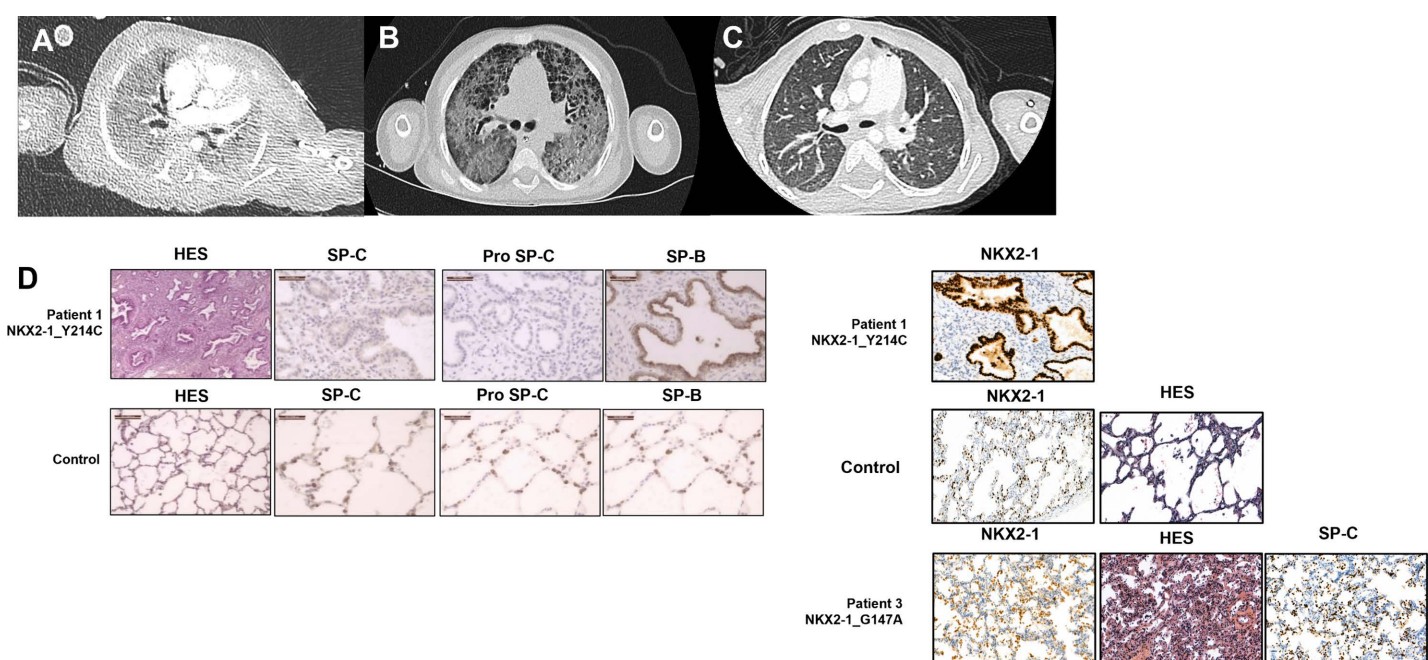

**Fig 1. Chest Computed Tomography of the three patients and lung biopsy patterns and expression of surfactant proteins.** (A) Patient 1: Diffuse dense and bilateral ground-glass opacities, low quality is due to ECMO support and cannula artefacts (B) Patient 2: Diffuse dense and bilateral ground-glass opacities with cystic lesions bronchial distortion in favour of pre-fibrotic lesions (C) Patient 3: Mild diffuse and bilateral ground-glass opacities (D) Lung biopsies analysis. HES staining at x20 magnification of patient 1 shows an acinar dysplasia pattern (rudimentary airways, columnar epithelium and prominent mesenchyme) compared to control (x10). Immunochemistry, at high magnification (x40) for patient 1 shows a weak staining for SP-B, a negative staining for Pro SP-C and SP-C, and a normal staining for SP-A. Although more cells are stained, they appear weaker than the cells of the control. A different software was used for patient 3 (HES, SP-C, SP-A and NKX2−1), and for the staining of NKX2−1 in patient 1; they were compared to a control. HES staining at x10 magnification of patient 3 shows hypertrophy of the small arterioles which were very tortuous and hyperplastic, no complex lesions of pulmonary arterial hypertension and no features of ILD. Immunohistochemistry shows normal staining for NKX2−1, SP-C and SP-A at x10 magnification. NKX2−1 staining for patient 1 was normal taking into account the cell hyperplasia at x20 magnification.

**Table 1. Clinical characteristics of the patients and in silico analysis of the variants.**

| | Patient 1 | Patient 2 | Patient 3 |
|---|---|---|---|
| *NKX2−1* variation (NM_003317.3) | c.641A>G p.(Tyr214Cys)) | c.493C>T p.(Arg165Trp) | c.440G>C p.(Gly147Ala) |
| *DNA* number | 19GM01101 | 8821GM000916 | 8821GM001014 |
| GnomAD v4.1.0 | Absent | Absent | 3/1601374 alleles |
| REVEL | Damaging (0.94) | Damaging (0.961) | Uncertain (0.421) |
| AlphaMissense | Likely Pathogenic (1.0) | Likely Pathogenic (1.0) | Likely benign (0.109) |
| Transmission | Absent from mother, father not tested | *de novo* | Inherited from father |
| Sex | Female | Male | Female |
| Age at onset | Birth | 2.5 years | 2 years |
| Brain phenotype | Neonatal Hypotonia | Hypotonia, developmental delay | No |
| Lung phenotype | Acinar Dysplasia | Interstitial lung disease | Pulmonary hypertension |
| Thyroid phenotype TSH us | Congenital hypothyroidism 308 mUI/L | Congenital hypothyroidism 40 mUI/L | Normal 2.59 mUI/L |
| Outcome | Died at 5 weeks | Died at 3 years | Died at 2.5 years |

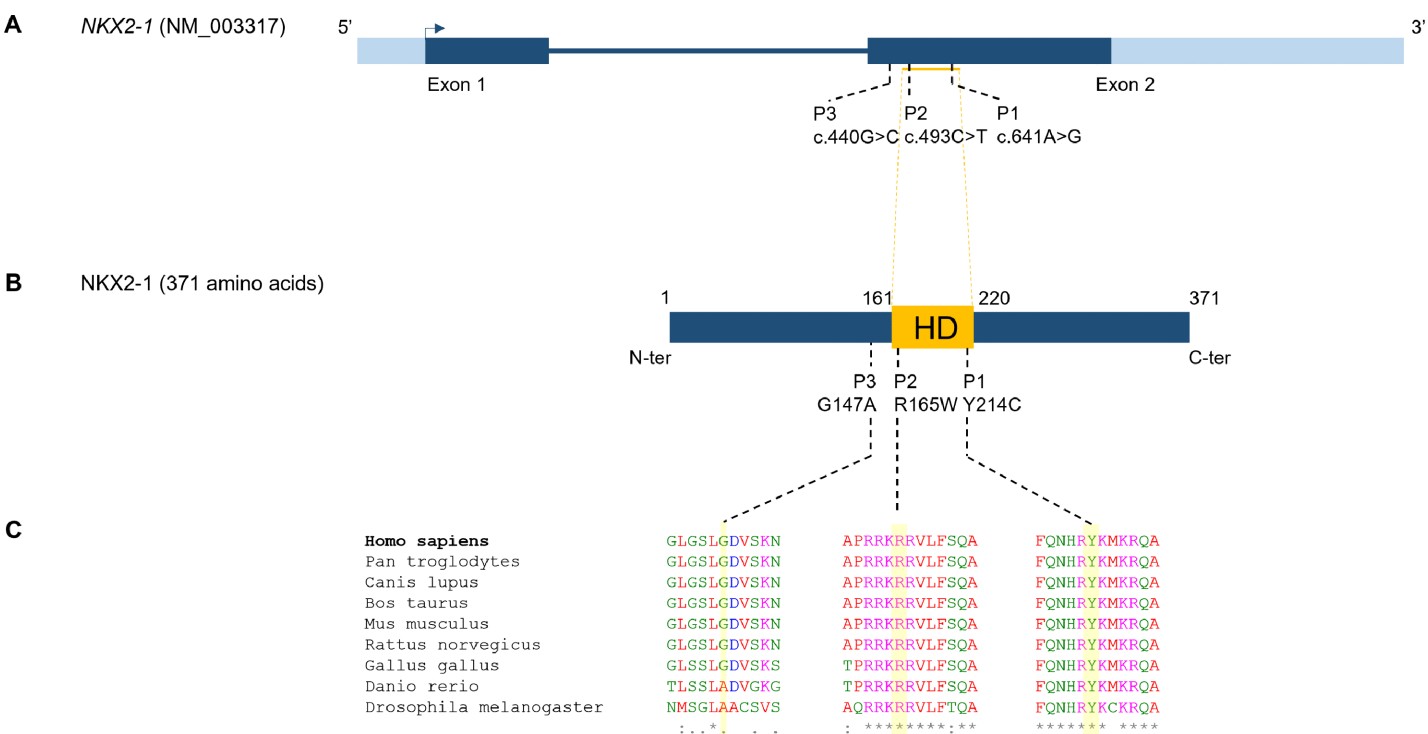

**Fig 2. Location and conservation of *NKX2−1* variants.** Localization of the variants on the gene (A) and on the protein (B). These two variants were located in the homeodomain. Amino acids R165 and Y214 are highly conserved in vertebrates (C).

and poor weight gain, an underlying lung disease was suspected rather than a post-infectious or ventilator-induced lung injury. He was also found to present slight hypotonia and neurodevelopmental delay, with no chorea and was also diagnosed with peripheral CH (Table 1), although neonatal screening did not reveal abnormal thyroid function. A chest CT

scan revealed architectural distortion with honeycombing, multiple ground-glass opacities, and diffuse alveolar consolidations (Fig 1B). Chest CT scan and thyroid echography revealed a normal-sized and normally located thyroid (S4 Fig). He was too unstable to undergo lung biopsy. Despite maximal management, he passed away at the age of 3 years. The parents refused lung transplantation and postmortem lung sampling. Targeted Sanger sequencing of *NKX2−1* revealed a *de novo* heterozygous missense *NKX2−1* variant, c.493C > T p.Arg165Trp (S3 Fig), located in the homeodomain of the protein, absent from GnomAD v4.1.0, affecting a conserved amino acid among vertebrates and predicted to be damaging (Alphamissense 1, REVEL 0.961) [4,15]. This variant was classified as pathogenic (Fig 2).

Patient 3's complete medical history was difficult to retrieve. She was born at 36 weeks gestational age (GA) in Turkey with intrauterine growth retardation and no respiratory distress at birth. She presented at 6 months of age with respiratory failure and deterioration of her general condition, with cyanosis and elevated liver enzymes. CT tomography revealed a pattern suggestive of ILD and dilation of the main pulmonary artery (Fig 1C). She developed PHT and required long-term supplemental oxygen therapy; she had no hypothyroidism or neurological symptoms. She died at 2.5 years of age following an exacerbation of refractory PHT, which led to cardiac arrest. Next Generation Sequencing (NGS) revealed a heterozygous missense variant in *NKX2−1,* c.440G > C p.(Gly147Ala), with no other variant or large deletion in *NKX2.1* or other ILD and PHT related genes. The variant was inherited from her asymptomatic father, and present in GnomAD v4.1.0 at extremely low frequencies (3/1601374 alleles) (S3 Fig). The amino acid Gly147 is partially conserved between vertebrates, but alanine is found at this position in fish and Drosophila melanogaster. It is located outside of the homeodomain and was classified as a variant of uncertain significance (VUS) (Table 1, Fig 2). The patient also had two siblings who died from respiratory failure after viral infection with no available DNA. Lung examination revealed hypertrophy of the small arterioles, which were very tortuous and hyperplastic. Interestingly, immunostaining for SP-C and NKX2−1 was normal compared to the control (Fig 1D). As no other molecular etiology was found in this family, despite evidence supporting the variant's benignity, we performed functional tests to evaluate it.

### NKX2−1 protein expression and subcellular localization

Protein expression was assessed 48 h after transfection. Western blot analysis showed that the expression levels of p.(Arg165Trp) and p.(Tyr214Cys) were significantly lower than those of NKX2–1_WT and p.(Gly147Ala) (Fig 3A). Subcellular protein localization was analyzed using immunofluorescence. NKX2–1_WT and its variants (p.(Gly147Ala), p.(Arg165Trp), and p.(Tyr214Cys)) were all localized in the nucleus of the transfected cells (Fig 3B). **Transactivation of NKX2−1 targeted promoters**

The ability of the three NKX2−1 variant proteins to transactivate specific lung promoters (*SFTPB* and *SFTPC*) was studied using a luciferase assay. The results showed that NKX2–1_WT and p.(Gly147Ala), induced transactivation of *SFTPB* and *SFTPC* promoters in HEK293T cells. The p.(Arg165Trp) variant induced increased transactivation of the *SFTPC* promoter, suggesting a gain of function and a non-significant decrease in the *SFTPB* promoter activity. In contrast, p.(Tyr214Cys) failed to transactivate the *SFTPB* and *SFTPC* promoters, suggesting a loss of function effect (Fig 4A).

The ability of the three variants to transactivate thyroid-specific promoters (*TG* and *TPO*) was also analyzed. NKX2–1_WT and p.(Gly147Ala) transactivated the *TG* and *TPO* promoters, whereas p.(Arg165Trp) was associated with lower transactivation of the *TG* and *TPO* promoters. The p.(Tyr214Cys) variant failed to transactivate the *TG* and *TPO* promoters. These results suggest a loss-of-function for the p.(Arg165Trp) and p.(Tyr214Cys) variants.

To mimic heterozygous alleles, as found in humans, the effect of the variants was tested by co-transfection with NKX2–1_WT, followed by luciferase assays (Fig 4B). Co-transfection of NKX2–1_WT with p.(Tyr214Cys) significantly impaired the transactivation of all promoters. Co-transfection of NKX2–1_WT with p.(Gly147Ala) did not impair the transactivation of any of the promoters. Co-transfection of NKX2–1_WT with p.(Arg165Trp) induced an increase in *SFTPC* promoter transactivation and a decrease in the transactivation of *SFTPB, TG* and *TPO* promoters.

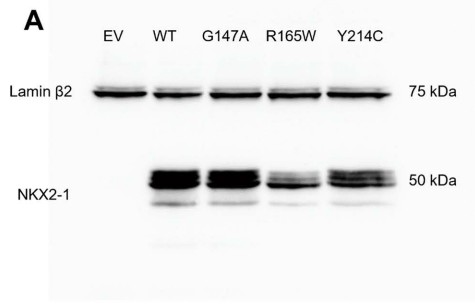

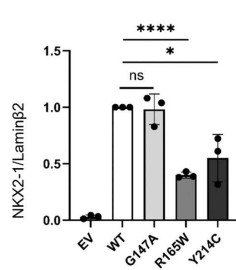

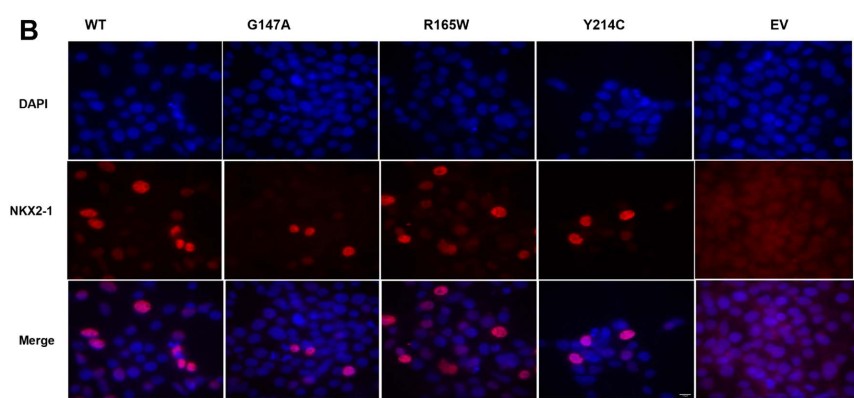

**Fig 3. Expression of the variants on Western Blot and Immunofluorescence in HEK293T.** (A) NKX2−1 protein expression in HEK293T cells transfected with WT, different mutant pNKX2−1 constructs or pCDNA3 empty vector (EV). R165W and Y214C induced a reduced expression of NKX2.1 whereas the G174A polymorphism is as expressed as the WT. Figures are representative of n = 3 replicates ns: non-significant, *: $p < 0.05$; **: $p < 0.01$, ***: $p < 0.001$. (B) Analysis of the subcellular localization of NKX2.1 examined with light microscopy. The 3 variants were localized in the nucleus, similarly to the WT. Nuclei were identified by DAPI staining. The scale bar indicates 10 µm. WT: NKX2−1 wild type. EV: Empty vector. G147A: p.(Gly147Ala). R165W: p.(Arg165Trp). Y214C: p.(Tyr214Cys). The WT was the NKX2–1_WT and the EV (Empty Vector) was the control.

Subsequently, we investigated the impact of the variants on the synergistic effect of NKX2−1 and PAX8 (Fig 5) on the transactivation of the *TG* and *TPO* promoters. PAX8_WT transactivated the *TG* and *TPO* promoters.

Co-transfection of NKX2–1_WT with PAX-8_WT displayed significantly enhanced synergy compared with individual transfections. It induced an 85% increase in *TG* promoter transactivation and a 63% increase in *TPO* promoter transactivation compared to NKX2–1_WT alone. Compared to PAX8 alone, it induced a 92% increase in *TG* promoter transactivation and a 53% increase in *TPO* promoter transactivation.

Co-transfection of PAX8_WT with p.(Arg165Trp) or p.(Tyr214Cys) induced significant impairment or abolished transactivation of the *TPO* and *TG* promoters, respectively, compared to co-transfection of PAX8_WT with NKX2–1_WT or p.(Gly147Ala).

The functional effects of the three variants are summarized in Table 2.

## Discussion

Our study assessed the pathogenicity of the p.(Tyr214Cys) variant and confirmed the pathogenicity of the p.(Arg165Trp) variant, both of which are located in the homeodomain of NKX2−1 containing the functional site and the nuclear localization signal of NKX2−1 [4,15–17].. It was also demonstrated that the p.(Gly147Ala) variant was indeed a benign polymorphism.

The expression of the p.(Gly147Ala) variant was conserved in western blot analysis, and the intra-nuclear sublocalization was analyzed by immunofluorescence. This variant did not impair the transactivation of any of the tested promoters. Even though Patient 3 presented with some patterns of ILD on the CT-scan, PHT, and a family history of unexplained death or severe respiratory disease, this variant was difficult to incriminate in her clinical picture and could finally be considered as a negative molecular control in our investigations.

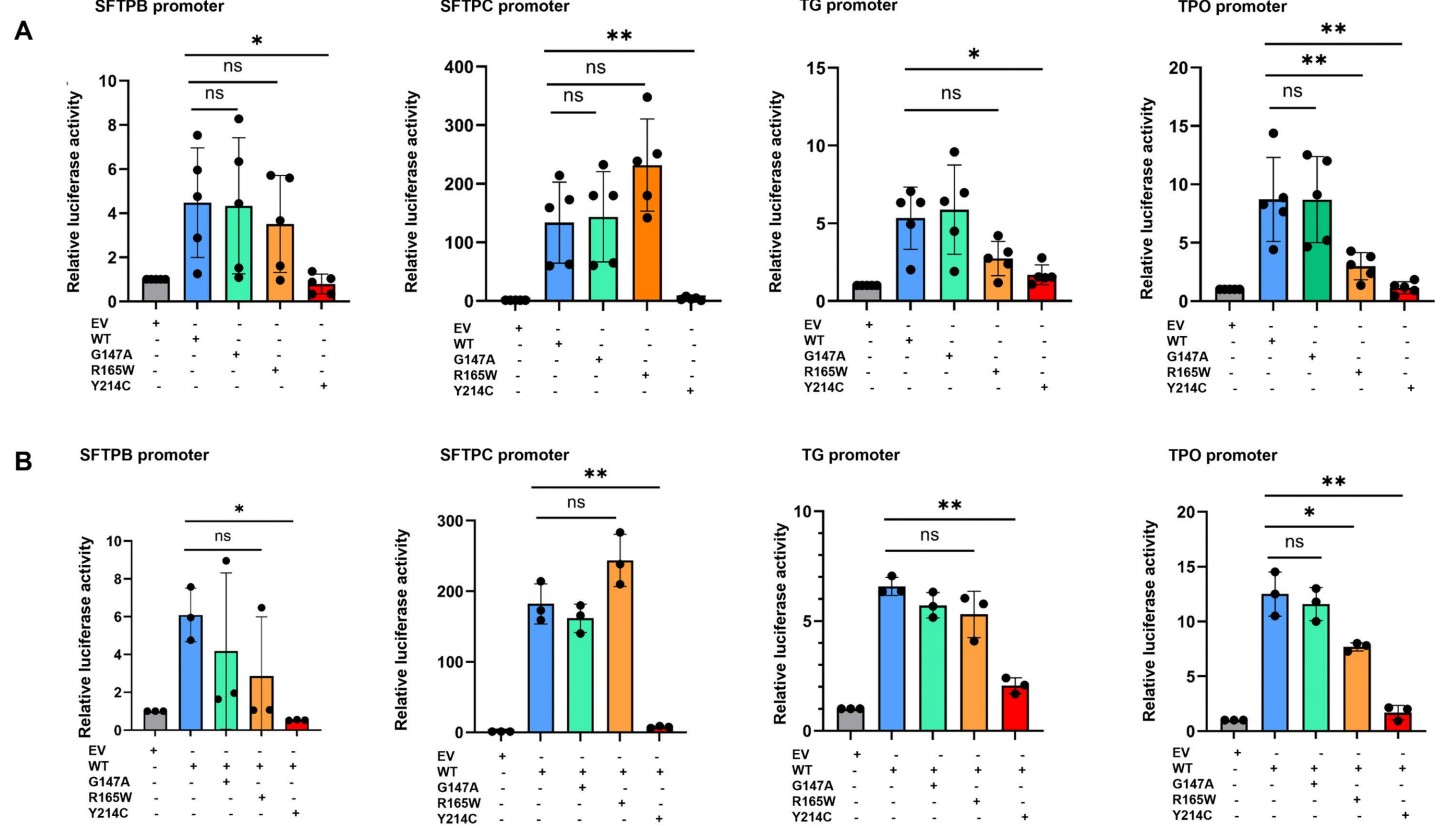

**Fig 4. Transactivation of specific promoters by luciferase assays.** Luciferase assays were performed 48 h after transfection of SFTPB (A), SFTPC (B), TG (C), and TPO (D) promoters in HEK293T cells without (A) or with (B) co-transfection of NKX2.1 WT with the variants. (A) R165W was associated with increased transactivation of the SFTPC promoter and decreased transactivation of TG and TPO; Y214C was associated with decreased transactivation of all tested promoters. Figures are representative of n = 5 replicates. (B) R165W co-transfected with WT impaired SFTPB and SFTPC promoter transactivation but not TG and TPO transactivation. Y214C co-transfected with WT impaired the transactivation of all tested promoters. Figures are representative of n = 3 replicates. TG: Thyroglobulin. TPO: Thyroperoxydase. EV: Empty Vector. WT: NKX2−1 wild type. G147A: p.(Gly147Ala). R165W: p.(Arg165Trp). Y214C: p.(Tyr214Cys). Results are expressed as the mean ± Standard Error of Mean (SEM). ns: non-significant, *: p < 0.05; **: p < 0.01, ***: p < 0.001. The WT was the NKX2−1_WT and the EV (Empty Vector) was the control.

Here, we demonstrated the pathogenicity of the p.(Tyr214Cys) variant and discussed its potential implication in diffuse abnormal lung development (acinar dysplasia) [13]. Western blot analysis revealed significantly decreased protein expression compared to that in the WT. Interestingly, NKX2−1 immunostaining appeared normal on the lung biopsy. This discrepancy may be due to the cellular hyperplasia observed in acinar dysplasia, which can produce apparently normal NKX2−1 staining even though expression is likely reduced. The reduced expression was observed in WB analysis performed with the mutated allele only, whereas the patient had one normal allele and one variant. Luciferase assays showed impaired transactivation of all tested promoters, and immunochemistry in her lung tissue showed the absence of SP-C and pro SP-C expression. Co-transfection of p.(Tyr214Cys) with the WT form of NKX2−1 and PAX8 also induced significantly reduced transactivation of all the tested promoters and impaired transactivation of *SFTPC* promoter. These results confirmed the potential haploinsufficiency of p.(Tyr214Cys). Moreover, Patient 1 exhibited the typical triad of symptoms associated with brain-lung-thyroid syndrome [1,4,16], presenting with respiratory disease, CH and neonatal hypotonia. The latter should be interpreted cautiously, as it was difficult to determine whether the hypotonia was due to her respiratory condition or part of the syndrome. Even though the final diagnosis was acinar dysplasia,

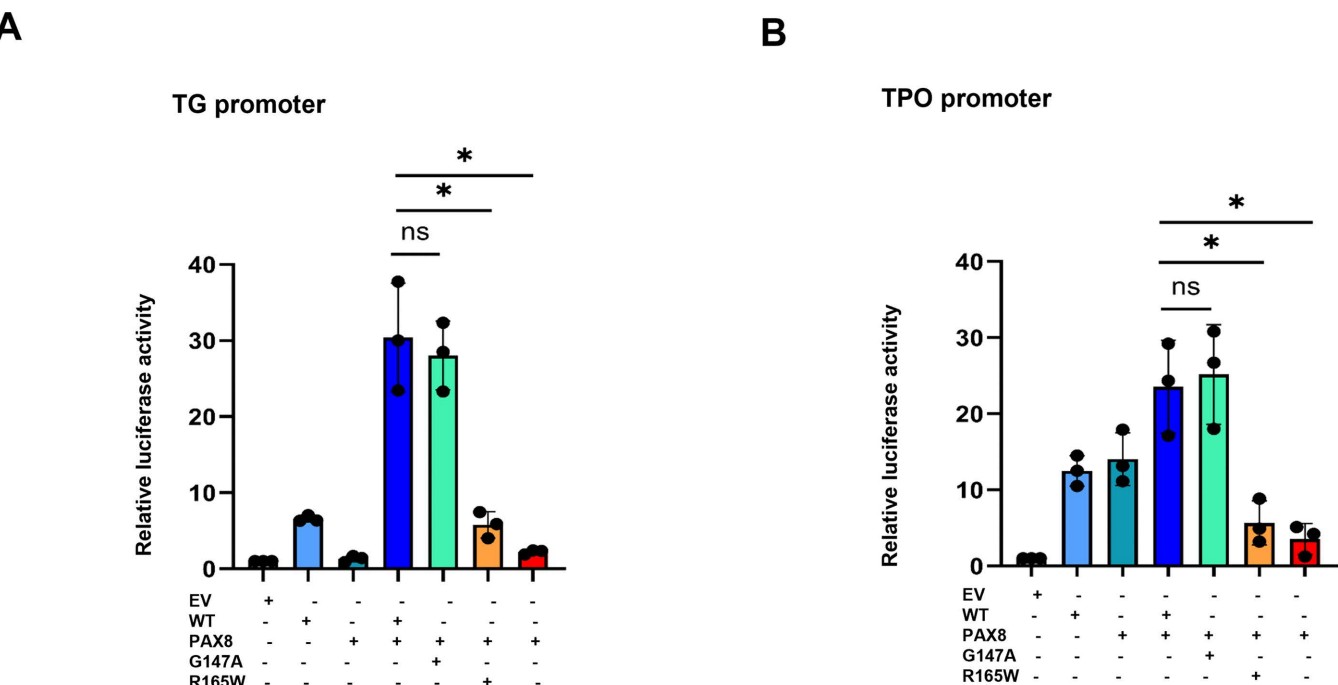

**Fig 5. Transactivation of specific promoters by luciferase assays after co-transfection of NKX2−1 and PAX8.** Luciferase assays were performed 48 h after transfection with the TG (A) and TPO (B) promoters. R165W and Y214C impaired the transactivation of TG and TPO promoters. TG: Thyroglobulin. TPO: Thyroperoxydase. EV: Empty Vector. WT: NKX2−1 wild type. G147A: p.(Gly147Ala). R165W: p.(Arg165Trp). Y214C: p.(Tyr214Cys). Results are expressed as the mean±Standard Error of Mean (SEM). ns: non-significant, *: p<0.05; **: p<0.01, ***: p<0.001. The WT was the NKX2−1_WT and the EV (Empty Vector) was the control.

**Table 2. Summary of the functional effects of the three variants on NKX2.1 pathogenic mechanisms.**

| Mechanisms | Patient 1 | Patient 2 | Patient 3 |
|---|---|---|---|
|  | p.(Tyr214Cys) | p.(Arg165Trp) | p.(Gly147Ala) |
| **Haploinsufficiency** (decreased expression on western blot analysis) | Yes | Yes | No |
| **Altered transactivation of the targeted promoters** (on luciferase assays) | Yes | Yes | No |
|  | SFTPC, SFTPB ⬇ | SFTPC ⬆ |  |
|  | TG, TPO ⬇ | TG, TPO ⬆ |  |
| **Altered nuclear translocation** (on immunofluorescence) | No | No | No |

her clinical presentation and chest imaging (x-ray and CT scan) were consistent with group A1 of the childhood ILD classification [18,19]. Unlike alveolar capillary dysplasia, which has been linked to variants in *FOXF1*, the genetic basis of acinar dysplasia remains poorly explained [18]. Although associations between mutations in *TBX4, FGF10, FGFR2* have been described [18,20,21], the exact molecular mechanisms are still not fully understood. *NKX2−1* is expressed during the early development of the lungs and is implicated in the branching of the lobar bronchi and in the later stages of lung development [12,13]. Our study sought to extend the understanding of the brain-lung-thyroid syndrome phenotype by including diffuse developmental disorders of the lung, thereby prompting inquiry into the mechanistic effect of NKX2−1 on pulmonary development.

We confirmed the pathogenicity of the p.(Arg165Trp) variant. Western blot analysis showed reduced protein expression compared to the WT, revealing a potential haploinsufficiency. However, luciferase assays revealed that the p.(Arg165Trp) variant protein was able to transactivate all the tested promoters. There was an increase in the transactivation of *SFTPC* promoter, confirming a dominant positive effect that was previously suggested [4,15]. The different effects of this variant were intriguing and led us to hypothesize that its pathogenicity could involve several pathways. Stuart et al. [22] demonstrated that NKX2−1 possesses multiple binding sites with unequal functional roles. This observation questioned if the sites were the p.(Arg165Trp) variant bound induced different effects on transactivation, depending on the involved promoter. Minoo et al [23] reported that NKX2−1 interact with FOXA1 to regulate SP-C protein expression, the p.(Arg165Trp) variant might have disrupted this interaction. Finally, Delestrain et al [24] found that some *SFTPC* variants led to the accumulation of pro SP-C in the endoplasmic reticulum leading to reduced amount of mature SP-C. Therefore, increased *SFTPC* promoter activation might result in the overproduction of specific isoforms, potentially causing pro-SPC accumulation in the endoplasmic reticulum and cell stress. Co-transfection of the p.(Arg165Trp)variant protein with *PAX8* drastically reduced *TG* and *TPO* promoters' transactivation. The dominant negative effects of other *NKX2−1* variants, including missense variants (L176V, R178P, K206E, and I207F) and frameshift variants (R165fs, L263fs, and P275fs), have previously been hypothesized in several studies [12,15,8, 25–31 ]. Therefore, the binding ability of NKX2−1 to DNA via its homeodomain can be impaired [16,25,32]. Guillot et al. [15] showed an abnormal NKX2−1 staining on the lung biopsy of the patient carrying the same p.(Arg165Trp) variant. The absence of lung specimens from patient 2 was a limitation to further elucidate this pathogenicity. Furthermore, to better characterize the pathogenicity of these two variants, more experiments would be relevant such as ChIP-qPCR or ChIP-seq. These experiments would be advantageous for elucidating the impact of the variant on its binding profile. Specifically, it would allow for the determination of whether the variant exhibits reduced or impaired binding at known NKX2−1 binding sites within its regulatory regions. Moreover, it would facilitate the identification of potential increase in, or aberrant localization of, variant binding throughout these regulatory regions [22].

To date, genotype-phenotype correlations for NKX2−1 variants are challenging given the variety of molecular variations and clinical presentations. Hamvas et al. [4] reported patients with large deletions encompassing NKX2−1 and adjacent genes and manifesting with an early-onset complete brain-lung-thyroid syndrome. Conversely, patients with whole-gene deletions exhibited milder pulmonary disease. The authors further hypothesized that certain variants may disrupt the interaction of NKX2.1 with its molecular partners, and that epigenetic factor could contribute to phenotypic variability.

Another example is in the study of Thorwarth et al. where the authors suggested that a single amino acid substitution within the homeodomain of NKX2−1 was less tolerated than other molecular variations [33]. More recently, Michel et al. [34] conducted a systematic review including 38 studies and 148 patients. Their findings highlighted a broad spectrum of symptoms and genetic variations, and pointed out that the three out of 115 patients whose disease was complicated by a lung cancer had nonsense variants.

In the present study, p.(Tyr214Cys) and p.(Arg165Trp), are missense variants located within the homeodomain of NKX2−1 (DNA-binding site). We hypothesized that impaired DNA binding might account for the severe effects on protein function observed with these variants, and further, that such impairment could serve as a predictive biomarker for disease severity.

## Materials and methods

### Ethics

The study was approved by the relevant ethics committees ("Comité de Protection des Personnes") under the number CPP20130604, and written informed consent was obtained from all the participants or legal representatives. Clinical information was collected from a legally authorized database (CNIL N°681248).

DNA samples from the three patients were available, with parental written consent obtained for DNA analysis by Sanger sequencing or NGS. Lung biopsies analysis was also performed with parental written consent. The study was approved by the relevant ethics committees.

## Molecular analysis

Genomic DNA was extracted from peripheral blood leukocytes of the probands and their parents.

Sanger sequencing targeting of *NKX2−1* was performed for patients 1 and 2 using an ABI 3730XL automated capillary DNA sequencer (Applied Biosystems). Sequencing data were then analyzed through Seqscape software.

For patient 3, next-generation sequencing (NGS) targeting all coding exons and intronic flanking regions of the following list of genes was performed: *ABCA3, COPA, CSF2RA, CSF2RB, FARSA, FARSB, FLNA, GATA2, MARS1, NKX2−1, OAS1, SFTPA1, SFTPA2, SFTPB, SFTPC, STING1*, and *TBX4* [35,36]. NGS was performed using KAPA Library Preparation and KAPA HyperCap Target Enrichment Probes (Roche Sequencing), and a Miseq (Illumina) platform, according to the manufacturer's instructions. Sequence data were analyzed by using an in-house bioinformatics pipeline. Sequence reads in fastq format were aligned to the reference human genome (hg19) with BWA and Bowtie2. Variant calling was performed with GATK and VarScan, using a threshold for the variant allele fraction (VAF) of 10%. Variant calls in VCF format were then annotated using Annovar. In the meantime, NGS for the following PHT-related genes was performed: *ACVRL1, BMP10, BMPR2, CAV1, EIF2AK4, ENG, EPHB4, FOXF1, GDF2, GLMN, KCNK3, KRIT1, PTEN, RASA1, SMAD4, SMAD9, SOX17, TBX4, TEK* (http://www.cgmc-psl.fr/spip.php?article45).

## Plasmids constructs

Reporter plasmids encompassing regulatory regions of *TG, TPO, SFTPB, and SFTPC*, as well as the expression plasmids *NKX2−1* and *PAX8* wild type (WT), were kindly provided by colleagues from INSERM U1016 and U938 teams [8,15].

*NKX2−1* (NM_003317.3) cDNA was cloned into pcDNA3.1 expression vector (Invitrogen), and site-directed mutagenesis was performed with the Q5 high-fidelity polymerase (New England Biolabs) to generate plasmids carrying the studied variations p.(Tyr214Cys), p.(Arg165Trp) and p.(Glyc147Ala). Plasmid construct sequences were confirmed by Sanger sequencing using the BigDye Terminator sequencing kit (Applied Biosystems) on an ABI 3730XL automated capillary DNA sequencer (Applied Biosystems). Plasmid maps are provided in the **Supplementary data** (S1 and S2 Files).

## Cell culture

HEK293T cells were cultured in Dulbecco's Modified Eagle Medium (DMEM) containing 4.5g/l glucose, supplemented with 10% fetal bovine serum (FBS) and 1% penicillin/streptomycin.

## Western blotting analysis

Western blot analyses were performed to verify the transfection and assess the expression of the different variants.

HEK293T cells were seeded at 2x10^5 cells/well in six-well plates 24h before transfection. Transfections were performed with 1 µg of expression plasmid (NKX2–1_WT, PAX8_WT, p.(Tyr214Cys), p.(Arg165Trp) and p.(Glyc147Ala)) using a 1:3 ratio of FUGENE HD® transfection reagent, according to the manufacturer's instructions. Fourty-eight hours post-transfection, cells were lysed with Complete 1X (protease inhibitor, Roche) and RIPA 1X (RIPA 5X: 250mM TrisHCl pH7.5, 750mM NaCl, 5% Triton, 5% Sodium Deoxycholate). Insoluble and soluble lysates were separated by centrifugation at 13,000×g for 10 min.

After dilution with 2X Laemmli buffer containing DTT, 15 µL of soluble lysate proteins was loaded under denaturing conditions and electrophoresed with Tris-Glycine SDS buffer. Gels were transferred onto a nitrocellulose membrane via semi-liquid transfer (Transblot, Bio-Rad). Membranes were blocked with 5% (w/v) milk in PBS 0.1% Tween-20 for one hour at room temperature (RT) and then incubated overnight with the following primary antibodies diluted in milk blocking buffer: anti-NKX2–1 (sc-53136, Santa Cruz Biotechnology, 1:500), anti-PAX8 (sc-81353, Santa Cruz Biotechnology, 1:1000) and anti-lamin β2 HRP (housekeeping protein, 1:1500). The secondary anti-mouse HRP was incubated for 1 h at RT (Sigma, A0545, 1:5000). Proteins were detected using Amersham ECL Select Western Blotting Detection Reagent (GE Healthcare), according to the manufacturer's recommendations.

## Immunohistochemistry

Immunochemistry was performed was performed to assess the function of NKX2−1 on patients' lung biopsies.

Lung biopsies from control, Patients 1 and Patient 3 were analyzed by hematoxylin and eosin staining (HES) and immunochemistry using an anti-SP-B antibody (Abcam ab271345, 1:1000), an anti-SP-C antibody (Santa Cruz Biotechnology Cat# sc-13979, RRID: AB_2185502, 1:1000), an anti-pro-SP-C (Abcam Cat# ab90716, RRID: AB_10674024, 1:1000), and an anti NKX2−1 (Abcam Cat # ab76013, RRID: AB_1310784).

## Immunofluorescence

Immunofluorescence was performed to verify the expression of the variants and the nuclear localization of NKX2−1 protein.

HEK293T cells were transfected with the expression plasmids (pNKX2–1_WT, p.(Tyr214Cys), p.(Arg165Trp) and p.(Glyc147Ala)) using a 1:3 ratio of FUGENE HD® (Promega, USA) transfection reagent. For immunofluorescence analysis, cells were fixed with 4% paraformaldehyde for 20 min, washed with PBS, and permeabilized with 0.3% triton x-100 in PBS containing 0.05% Tween-20 for 20 min. Nonspecific binding sites were blocked with 2% BSA for one hour, then incubated with a primary NKX2−1 antibody (sc 53136, 1:300) overnight at 4°C, and with a secondary fluorescently labeled antibody (Alexa Fluor™, Invitrogen, 1:1000) for 1 h at RT. and Finally, cells were mounted with ProLong® antifade reagent with DAPI (Invitrogen).

## Luciferase assays

Luciferase assays were performed to test the transactivation of NKX2−1 targeted promoters by the different variants.

HEK293T cells were plated at 1x10^5 cells/well in twelve-well plates 24h before transfection. Cells were co-transfected with 500 ng of reporter plasmids (*SFTPB, SFTP-C, TG, or TPO)*, 250 ng of pcDNA3, and NKX2−1 WT or variants, along with 10 ng of Renilla luciferase expression plasmid using 1:3 ratio of FUGENE HD® (Promega) transfection reagent. PAX8-NKX2–1 co-transfection experiments were performed using 125 ng of PAX8 and 125 ng of NKX2−1. Fourty-eight hours post-transfection, the cells were harvested for dual-luciferase reporter assays (Promega) using a TriStar LB 941 Multimode Microplate Reader (Berthold Technologies). Firefly luciferase enzyme activity was normalized to Renilla luciferase activity.

## Statistics

Statistical analyses were performed using non-parametric Kruskal-Wallis tests with post-hoc tests in GraphPad Prism v10.0. Data are representative of three to five independent experiments. Statistical significance was set at $p$ value ≤ 0.05.

## Conclusion

This study describes the pathogenicity of two variants located within the homeodomain of *NKX2−1.* These variants, both associated with lethal lung phenotypes, altered the transactivation of lung and thyroid promoters. Although a correlation was observed between functional impairment and clinical severity, a direct causal relationship remains to be definitively established. Further investigation is also warranted to characterize the impact of *NKX2−1* variants located outside the homeodomain.

## Supporting information

**S1 Fig. Chest X-ray of patient 1.** Diffuse dense and bilateral ground-glass opacities.
(TIF)

**S2 Fig. Thyroid Ultra Sound of patient 1.** Normal structure and location.
(TIF)

**S3 Fig. Sanger Sequencing results of the 3 patients.**
(TIF)

**S4 Fig. Thyroid Ultra Sound of patient 2.** Normal structure and location.
(TIF)

**S5 Fig. Raw Western Blots.**
(PDF)

**S1 File. Plasmids maps with the variants.**
(RTF)

**S2 File. Plasmid map of the wild type.**
(DOC)

**S1 Raw Data. Soreze PONE-D-25-14519 Raw data luciferase.**
(XLSX)

## Acknowledgments

We thank the patients and their families for their assistance. We thank the French National Networks for Rare Lung Diseases, RespiRare, and RespiFIL. The ILD cohort was developed in collaboration with the Rare Disease Cohort (RaDiCo)-ILD project (ANR-10-COHO-0003) and ERS Clinical Research Collaboration for ChILDEU.

## Author contributions

**Conceptualization:** Yohan David SOREZE.

**Data curation:** Aurore Coulomb-l'Herminé.

**Formal analysis:** Yohan David SOREZE.

**Methodology:** Tifenn Desroziers-Louedec, Farah Diab, Aphrodite Daskalopoulou, Valérie Nau, Camille Louvrier.

**Project administration:** Nadia Nathan.

**Supervision:** Farah Diab, Aphrodite Daskalopoulou, Marie Legendre, Sonia-Athina Karabina, Véronique Houdouin, Camille Louvrier, Nadia Nathan.

**Validation:** Aurore Carré, Farah Diab, Aphrodite Daskalopoulou, Marie Legendre, Sonia-Athina Karabina, Véronique Houdouin, Camille Louvrier, Nadia Nathan.

**Writing – original draft:** Yohan David SOREZE.

**Writing – review & editing:** Tifenn Desroziers-Louedec, Aurore Carré, Farah Diab, Aphrodite Daskalopoulou, Julie Starck, Marie Legendre, Sonia-Athina Karabina, Aurore Coulomb-l'Herminé, Camille Louvrier, Nadia Nathan.

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
