## [Decision Letter · Decision Letter 0]

30 Jul 2025

Dear Dr. SOREZE,

Thank you for submitting your manuscript to PLOS ONE. After careful consideration, we feel that it has merit but does not fully meet PLOS ONE’s publication criteria as it currently stands. Therefore, we invite you to submit a revised version of the manuscript that comprehensively addresses the points raised by all three reviewers.

We look forward to receiving your revised manuscript.

Kind regards,

Andre van Wijnen, PhD

Academic Editor

PLOS ONE

Journal Requirements:

“NN:

Grants or contracts from any entity

2024: CORTICONEHI: Clinical trial: Efficacy of methylprednisolone pulses in

neuroendocrine cells hyperplasia of infancy: an early phase study.

2023: Million Dollar Bike Ride project for Neuroendocrine Cell Hyperplasia of Infancy:

Genetic basis of neuroendocrine cell hyperplasia of infancy

2022: Chancellerie des Universités: Legs Poix, Molecular and phenotypic

characterization of interstitial lung disease n°2022000594

2022: RespiFIL grant for the development of an e-learning module for CT-scan in

childhood interstitial lung diseases (15 000 €)

2022: RespiFIL grant for the development of an online platform for the collection of

quality of life and transition questionnaires in rare lung disease (15 000 €)

Payment or honoraria for lectures, presentations, speakers bureaus, manuscript writing

or educational events

2022: La lettre du Pneumologue

Support for attending meetings and/or travel: 2023: ERS Travel grant

Leadership or fiduciary role in other board, society, committee or advocacy group, paid

or unpaid

2021-2025 Head of the ERS Clinical research collaboration for childhood ILD (CRC[1]chILDEU)

2017-2023 Treasurer of the Société française de pédiatrie (SFP)

2023- Treasurer and Scientific committee of the Société de Pneumologie Pédiatrique

et d’Allergologie (SP2A)

2023- Scientific and Scientific committee of the Société de Pneumologie de Langue

Française (SPLF)

AC:

All support for the present manuscript

Supply of vectors”

Please include your updated Competing Interests statement in your cover letter; we will change the online submission form on your behalf.”

Reviewers' comments:

Reviewer's Responses to Questions

**Comments to the Author**

1. Is the manuscript technically sound, and do the data support the conclusions?

Reviewer #1: No

Reviewer #2: Partly

Reviewer #3: Yes

2. Has the statistical analysis been performed appropriately and rigorously?

Reviewer #1: N/A

Reviewer #2: No

Reviewer #3: Yes

3. Have the authors made all data underlying the findings in their manuscript fully available?

Reviewer #1: Yes

Reviewer #2: Yes

Reviewer #3: Yes

4. Is the manuscript presented in an intelligible fashion and written in standard English?

Reviewer #1: Yes

Reviewer #2: Yes

Reviewer #3: Yes

Reviewer #1: The authors present 3 cases of "ultra-severe forms of childhood interstitial lung disease" and a functional analysis of 3 NKX2.1 variants identified in the subjects. Actually, only one variant (#1) is novel and meets the aims in the title. The phenotype-genotype correlation is unclear for cases #2 and 3. The variants’ functional testing is not novel.

There are many grammatical issues and the language should be reviewed by a native English speaker.

Major concerns:

Case reports

Case 1

- Clinical presentation seems typical of Persistent pulmonary hypertension of the newborn (PPHN). The chest CT is of poor quality and cannot support a diagnosis of interstitial lung disease, but rather shows diffusely decreased aeration.

- The correct histological phenotype is acinar dysplasia, which is a developmental lung disease, not ILD.

- What was the purpose of surfactant protein immunostaining, and what is the relevance of the results in the context of the paper? In figure 1D legend, the authors report weak SP-B staining, which seems the opposite on the corresponding image. Considerations on protein expression in lung tissue should be supported by Western blot.

- How was genetic testing done? Which genes were tested? Was the variant inherited or de novo? Was a molecular caryotype performed (as TTF-1 deficiency can be caused by CNVs)?

- Was the thyroid gland absent, ectopic, or dysplastic?

- Was there a neurological phenotype?

Case 2

- The clinical history is difficult to follow. When was the chest CT performed in the course? Was it interpreted as post-RSV changes, VILI, or ILD, or a combination? Genetics: same questions as above.

Case 3

- The clinical history is unclear. It is compatible with ILD. The chest CT seems indicative of PAH, not typical of ILD. Please provide a more accurate description. Same question as above for thyroid and brain.

- The relevance of this case is questionable as molecular diagnosis remains unknown. Again, which genetic tests were performed?

Functional analysis of the three variants

This section is more relevant and novel. However, the authors should explain the rationale for the tests performed. NKX2.1 deficiency mechanisms include defective nuclear translocation (Peca et al. Respiratory Research 2011, 12:115), haploinsufficiency (decreased transcription, decreased translation), or functional defects (lack of promoter transactivation in target genes) among others. This should be discussed in the manuscript. A table summarizing the findings would be welcome.

- On figure 3B, the control has poor and diffuse TF-1 staining, and doesn’t demonstrate nuclear staining, contrarily to the legend.

- For the Y214C variant, do your results suggest a combined mechanism of decreased expression (haploinsufficiency) and loss-of-function (impaired transactivation)? Again, a table comparing the different effects of the 3 variants would help.

- R165W DNA-binding analysis is not novel (ref 14). The functional tests presented here are similar, and just confirm the published findings, i.e. domaicominant-positive effect in the lung, not in the thyroid. However, the authors report also partial haploinsufficiency. How do you reconciliate these findings in order to explain the clinical impact? Interestingly the dominant-positive effect is seen only on the SFTP-C promoter, not on SFTP-B. Can it depend on opposite effects on the promoter of SFTP-B or other surfactant-related proteins regulated by NKX2.1, such as ABCA3?

Minor concerns:

Typically, NKX2.1, SFTP-B, SFTP-C designate the gene, whereas TTF-1, SP-B, SP-C designates the protein. I suggest to adjust the whole text for clarity. Genes should be in italic.

Variants should be reported in the HGVS Nomenclature throughout the text; p.(Tyr214Cys) c.641A>G

Line 55-56: that are associated with ultra-severe forms of childhood interstitial lung disease (chILD) in infants: not a true statement

Line 114: The patient was diagnosed with peripheral CH: when? Was he asymptomatic until age 2.5?

Line 115: architectural destruction with

Line 123: weeks gestation

Line 127-9: She developed PHT and required long-term supplemental oxygen therapy;

She died of refractory cardiac arrest… pulmonary vasodilators? died of right heart failure? These are not appropriate clinical descriptions.

Line 198-204: this paragraph is not pertinent to the discussion

Line 217-220: this paragraph is more background, not discussion

Reviewer #2: This study by Yohan Soreze and colleagues investigates three NKX2-1 variants (Y214C, R165W, G147A) associated with severe childhood interstitial lung disease. Through functional assays, the authors demonstrate that Y214C is a loss-of-function mutation with no promoter activation, R165W shows mixed effects, and G147A appears benign. These findings support the pathogenicity of Y214C and R165W, with Y214C representing the most severe phenotype reported to date. Although the study focuses on only three variants and uses HEK293 cells, the authors enhance the physiological relevance by co-transfecting transcriptional partners.

The manuscript is generally well written and easy to follow. However, I have several questions and suggestions that I hope will help improve the clarity and scientific rigor of the work.

Major Comments

1. Statistics and Replicates

• Most assays appear to be performed in triplicates (n=3), which is insufficient to reliably assume normality or robustly estimate variance.

• Normality tests are not reported, and multiple comparisons are conducted without appropriate correction.

• Figure 4A and 4B for TG promoter seem to be the same plot, I am sure the authors placed it by mistake.

Recommendations:

• Increase the number of biological replicates to strengthen the statistical power; or

• If no additional experiments are planned, consider using non-parametric statistical tests (e.g., Kruskal–Wallis) with appropriate corrections for multiple comparisons (e.g., Dunn’s or FDR correction).

• Check Figure 4 plots for TG promoter and put the correct plots.

2. Methodological Detail and Reproducibility

• Key details are missing from the Methods section, making replication difficult.

Recommendations:

• Clearly describe the reporter plasmids: specify promoter lengths, vector backbones, and whether they are commercially or publicly available (e.g., Addgene codes).

• If not available via repositories, consider uploading plasmid maps to the supplementary material.

• Consider including a known pathogenic or benign NKX2-1 variant as an internal control in the luciferase assays to contextualize the functional results.

3. Interpretation of R165W Functional Effects

• Please elaborate on the R165W variant. Where is it located within the NKX2-1 structure? Does the affected residue have any known structural or functional importance?

• The gain-of-function effect seen with only one promoter is very intriguing and a few extra experiments could be interesting.

Suggestion:

While luciferase assays are useful for assessing promoter activity, they are limited in their scope. Additional experiments, such as ChIP-qPCR or ChIP-seq at relevant promoter sites, could provide stronger mechanistic insight into the variant-specific effects. I understand this is a tedious technique, and this is just a suggestion for the future.

4. Interpretation of Clinical Data

• The conclusion stating that “impairment of promoter transactivation led to acinar dysplasia…” may be overstated, given the current functional evidence.

Suggestion:

Rephrase the conclusion to reflect that while functional impairment correlates with clinical severity, a direct causal link has not been definitively established.

Minor Comments

• Terminology: The term “next-generation sequencing (NGS)” is somewhat outdated. Please consider using “high-throughput sequencing” instead.

• Sequencing Methods: Clarify the type of sequencing used (e.g., exome, panel, genome). Describe how variants were called and filtered. Providing raw variant calling files as supplementary material would enhance transparency and allow reproducibility.

• Variant Nomenclature: Please follow HGVS recommendations. Use three-letter amino acid codes where appropriate (e.g., p.Tyr214Cys instead of Y214C). Tools like Mutalyzer can help the authors with this.

• Line 200 – Acinar Dysplasia Associations: The association between acinar dysplasia and mutations in TBX4, FGF10, and FGFR2 is now well supported across multiple studies. While molecular mechanisms remain incompletely understood, newer model systems (e.g., lung organoids) are likely to help us understand these pathways in the context of AD.

• Align well Figure 4 dashes.

Reviewer #3: This article describes the pathogenicity of three variants of the NKX2-1 gene associated with lethal forms of childhood interstitial lung disease : two likely pathogenic and located in the homeodomain of the NKX2-1 protein (p.Tyr214Cys and p.Arg165Trp) and one likely benign (p.Gly147Ala).

The authors report clinical phenotypes, chest CT findings and lung histology (when available). Functional analyses of NKX2-1 variants were performed in HEK293T cells, assessing protein expression, subcellular localization and transactivation of lung and thyroid specific promoters with or without PAX8 cotransfection. Two patients carrying Y214C and R165W variants exhibited brain-lung-thyroid syndrome and severe interstitial lung disease, with one showing atypical acinar dysplasia. In contrast, the patient with the G147A variant presented isolated lung disease, primarily characterized by pulmonary hypertension. In vitro functional studies confirmed the pathogenicity of two variants, Y214C and R165W, while the G147A variant showed no deleterious effect. Western-blot analysis demonstrated reduced protein level for Y214C and R165W with no change in their cellular localization. The R165W variant increased transactivation of the SFTPC promoter, caused a non significant decrease in the SFTPB promoter activity, and significantly reduced transactivation of the TG and TPO promoters, even with PAX 8 cotransfection. The Y214C variant significantly impaired transactivation of all tested promoters.

Surfactant disorders caused by NKX2-1 mutations are responsible for rare lung disease. To date, few functional studies have assessed the impact of NKX2-1 variants, making this manuscript particularly valuable for correlating genotype and phenotype. The methods used are appropriate to investigate the functional consequences of these variants.

However, the manuscript requires some minors revisions :

• Page 4, ligne 87 : « Heterozygous pathogenic variants of NKX2-1 are associated with brain-lung-thyroid syndrome » : 56% of patients carrying NKX2-1 mutations present with brain-lung-thyroid syndrome while the others exhibit mono or bi-organ involvement, as observed in patient 3 described in this study. It would therefore be relevant to specify this point in the manuscript.

• Page 6 line 114 : « The patient was diagnosed with peripheral CH » The abbreviations « CH » should be defined when first mentionned in the manuscript.

• Page 6 line 142 : There appears to be typographical error in the variant name « Y214C », where the « C » (for cysteine) is missing.

• Page 6 line 142 : « Western blot analysis showed that

R165W and Y214 expression levels were significantly lower than those in WT and G147A » Although the mutated NKX2-1 protein shows reduced expression in HEK293 cells compared to the WT, NKX2-1 expression was reported as normal in the lung biopsy of patient 1. The authors should discuss this apparent discrepancy.

• The nomenclature used to describe the protein variants is inconsistent throughout the manuscript. The authors should standardize the naming convention to ensure clarity and coherence. For example, the same variant is referred to as NKX2-1_p.Y214C (page 7 line 157), NKX2-1_Y214C ( page 7 line 162), Y214C (page 7 line 145) or p.Tyr214Cys (Page 5 line 106).

• Figure 3 : The authors should standardize the legends between the western-blot and Immunofluorescence panels to facilitate interpretation. For example, does « Empty vector » in the western-blot correspond to « Control » in the immunofluorescence ? Does « NKX2-1 » in the immunofluorescence refer to the « WT » condition in the western-blot ? In addition, the nomenclature used to describe the protein variants in the figure should be consistent with that used in the main text.

• In Page 9, line 201 : The statement « Our study contributes to the involvement of NKX2-1 in lung development, endorsing an important role between the pseudoglandular (8WG) and canalicular stages (26 WG) » would benefit from further support. The authors are encouraged to provide additional data, references, or justification to substantiate this claim.

• In page 9, line 208 : « There was a significant increase in the transactivation of SFTPC promoter, confirming a dominant positive effect that was previously suggested » The authors demonstrate that the R165W variant enfances transactivation of the SFTPC promoter. However, the potential pathogenic mechanism underlying this effect remains unexplored. They should discuss how increased SFTPC expression possibly through ER stress, disrupted alveolar type II cell homeostasis or inflammatory signaling might contribute to the development of interstitial lung disease.

• In Page 10, line 225 : « The two pathogenic variants reported here, Y214C and R165W, are single nucleotide substitutions resulting in missense variants in the homeodomain of NKX2-1 (DNA-binding site). This binding impairment might explain the highly deleterious consequences on protein function and could be used as a predictive factor of severity ». The authors should reconsider the phrasing regarding the genotype-phenotype association, as this correlation is not consistently observed in all cases. Notably, whole gene deletions have been reported to cause milder disease phenotypes. It would be helpful to clarify whether the severity of the disease is influenced solely by the type of mutation or also by its location within the gene. Furthermore, the authors are encouraged to discuss additional factors that may contribute to the variability in disease severity, such as modifier genes or epigenetic mechanisms.

**Do you want your identity to be public for this peer review?** For information about this choice, including consent withdrawal, please see our Privacy Policy

Reviewer #1: No

Reviewer #2: No

Reviewer #3: No

---

## [Author Response · Author response to Decision Letter 1]

3 Nov 2025

Journal Requirements:

“NN:

Grants or contracts from any entity

2024: CORTICONEHI: Clinical trial: Efficacy of methylprednisolone pulses in

neuroendocrine cells hyperplasia of infancy: an early phase study.

2023: Million Dollar Bike Ride project for Neuroendocrine Cell Hyperplasia of Infancy:

Genetic basis of neuroendocrine cell hyperplasia of infancy

2022: Chancellerie des Universités: Legs Poix, Molecular and phenotypic

characterization of interstitial lung disease n°2022000594

2022: RespiFIL grant for the development of an e-learning module for CT-scan in

childhood interstitial lung diseases (15 000 €)

2022: RespiFIL grant for the development of an online platform for the collection of

quality of life and transition questionnaires in rare lung disease (15 000 €)

Payment or honoraria for lectures, presentations, speakers bureaus, manuscript writing

or educational events

2022: La lettre du Pneumologue

Support for attending meetings and/or travel: 2023: ERS Travel grant

Leadership or fiduciary role in other board, society, committee or advocacy group, paid

or unpaid

2021-2025 Head of the ERS Clinical research collaboration for childhood ILD (CRC[1]chILDEU)

2017-2023 Treasurer of the Société française de pédiatrie (SFP)

2023- Treasurer and Scientific committee of the Société de Pneumologie Pédiatrique

et d’Allergologie (SP2A)

2023- Scientific and Scientific committee of the Société de Pneumologie de Langue

Française (SPLF)

AC:

All support for the present manuscript

Supply of vectors”

Please include your updated Competing Interests statement in your cover letter; we will change the online submission form on your behalf.”

All these data are submitted in the manuscript.

All data are available on request to the authors.

Reviewers' comments:

Reviewer's Responses to Questions

5. Review Comments to the Author

Reviewer #1: The authors present 3 cases of "ultra-severe forms of childhood interstitial lung disease" and a functional analysis of 3 NKX2.1 variants identified in the subjects. Actually, only one variant (#1) is novel and meets the aims in the title. The phenotype-genotype correlation is unclear for cases #2 and 3. The variants’ functional testing is not novel.

There are many grammatical issues and the language should be reviewed by a native English speaker.

Major concerns:

Case reports

Case 1

- Clinical presentation seems typical of Persistent pulmonary hypertension of the newborn (PPHN). The chest CT is of poor quality and cannot support a diagnosis of interstitial lung disease but rather shows diffusely decreased aeration.

We thank the reviewer for this comment. Chest CT of Patient 1 was performed while the patient was under ECMO support, what explains its poor quality. The CT chest was reviewed with an expert pediatric radiologist, and diffuse ground glass opacities (GGO) were attested. Chest Xray being generally less impacted by the presence of the cannulas than CT, we added in the Supplemental Materials a chest X-ray of Patient 1 showing GGO as well. This info was added lines 106-108 and in Figure 1A’s legend.

- The correct histological phenotype is acinar dysplasia, which is a developmental lung disease, not ILD.

We thank the reviewer for this comment. Acinar dysplasia is indeed a diffuse developmental lung disease. However, because it can manifest as an ILD pattern, with GGO, as in Patient 1, this disease is included in most childhood Interstitial lung diseases (chILD), as in Griese M. et al, Eur Respir Rev, 2018 (Groupe A1). We added this info in the discussion section lines 235-237.

- What was the purpose of surfactant protein immunostaining, and what is the relevance of the results in the context of the paper? In figure 1D legend, the authors report weak SP-B staining, which seems the opposite on the corresponding image. Considerations on protein expression in lung tissue should be supported by Western blot.

We thank the reviewer for this comment and this question. Surfactant protein immunostaining was performed to assess the function of NKX2-1 in patient lung biopsies, since NKX2-1 is a transcription factor of surfactant proteins. The weak staining of SP-B may indeed be difficult to observe due to the hyperplasia of type 2 alveolar cells. Although more cells are stained, they appear weaker than the cells of the control. This information has been added to Figure 1’s legend. Unfortunately, due to technical issues, we were not able to perform western blot in lung tissue from patients. This was added to the manuscript lines 113-116.

- How was genetic testing done? Which genes were tested? Was the variant inherited or de novo? Was a molecular caryotype performed (as TTF-1 deficiency can be caused by CNVs)?

We thank the reviewer for these questions. Genomic DNA was extracted from peripheral blood leukocytes of the probands and their parents. Sanger sequencing targeting NKX2-1 was performed for patients 1 and 2. It was performed on an ABI 3730XL automated capillary DNA sequencer (Applied Biosystems). Sequencing data were then analyzed through the Seqscape software.

In the meantime, NGS for the following PHT-related genes was also performed: ACVRL1, BMP10, BMPR2, CAV1, EIF2AK4, ENG, EPHB4, FOXF1, GDF2, GLMN, KCNK3, KRIT1, PTEN, RASA1, SMAD4, SMAD9, SOX17, TBX4, TEK (http://www.cgmc-psl.fr/spip.php?article45). This was added in the manuscript, in the methods section, lines 299-314.

As reported in PMID: 38035569 (doi: 10.1159/000534076), the variant was not found in the mother. The father was asymptomatic but did not consent to the DNA analysis. This was added in Table 1.

No karyotype or SNP arrays were performed in the three patients. CNVs can be detected using NGS targeted sequencing. For Patients 1 and 2, whose phenotypes were highly suggestive of brain-lung-thyroid syndrome, Sanger sequencing identified likely pathogenic variants, and confirmed the diagnosis. The heterozygosity of the variants confirmed the presence of two alleles, ruling out the possibility of a large deletion affecting the entire NKX2-1 gene. For Patient 3, who carried a variant of uncertain significance, no deletion of the NKX2-1 gene was identified by NGS. This was added in the manuscript, line 155-156.

- Was the thyroid gland absent, ectopic, or dysplastic?

We thank the reviewer for this question. The thyroid gland was present and in normal position. This information was added in the manuscript lines 107-108. We added in the Supplemental data some thyroid imaging of the Patient 1.

- Was there a neurological phenotype?

We thank the reviewer for this question. As reported in PMID: 38035569 (doi: 10.1159/000534076), the patient presented at birth with severe hypotonia and respiratory failure. However, it was impossible to discriminate if this hypotonia was related to her respiratory condition or to NKX2-1 disease. This was added in the manuscript line and line 109-110, 231-235 (discussion and results + Table 1).

Case 2

- The clinical history is difficult to follow. When was the chest CT performed in the course? Was it interpreted as post-RSV changes, VILI, or ILD, or a combination? Genetics: same questions as above.

We thank the reviewer for these questions. Questions related to molecular aspects for patient 2 were answered ahead.

The patient was unknown from our hospital before the age of 2 years old when he presented with severe respiratory distress in a context of RSV infection. Because of the uncommon evolution and clinical signs of respiratory insufficiency such as digital clubbing and poor weight gain, a chest CT scan was performed leading to the diagnosis of ILD. Thus, this CT scan was interpreted as ILD, not post RSV changes nor VILI. Thereafter, congenital hypothyroidism was diagnosed and NGS analysis identified the de novo missense heterozygous NKX2-1 variant c.493C>T p.(Arg165Trp).

The patient had congenital hypothyroidism. The thyroid gland was present and in normal position. We added in the supplemental data some thyroid imaging of patient 2. The patient had a slight hypotonia and neurodevelopmental delay but no chorea. The patient history was completed in the manuscript line 130-136

Case 3

- The clinical history is unclear. It is compatible with ILD. The chest CT seems indicative of PAH, not typical of ILD. Please provide a more accurate description. Same question as above for thyroid and brain.

We thank the reviewer for these comments. Patient 3 was not born in France, so the beginning of her medical history was difficult to retrieve. She presented with symptoms of respiratory failure at the age of 6 months old. Even though there were patterns of ILD on her chest CT scan, her medical history and lung biopsy were compatible with pulmonary hypertension. She didn’t have hypothyroidism and had no neurological impairment. We clarified the clinical history in the manuscript lines 143-150.

- The relevance of this case is questionable as molecular diagnosis remains unknown. Again, which genetic tests were performed?

We thank the reviewer for these questions. For patient 3, NGS targeting all the coding exons and the intronic flanking regions of the following list of genes was performed: ABCA3, COPA, CSF2RA, CSF2RB, FARSA, FARSB, FLNA, GATA2, MARS1, NKX2-1, OAS1, SFTPA1, SFTPA2, SFTPB, SFTPC, STING1, and TBX4. NGS was performed using KAPA Library Preparation and KAPA HyperCap Target Enrichment Probes (Roche Sequencing), and using a Miseq (Illumina) platform according to the manufacturer’s instructions. Sequence data were analyzed by using an in-house bioinformatics pipeline. Sequence reads in fastq format were aligned to the reference human genome (hg19) with BWA and Bowtie 2. Variant calling was performed with GATK and VarScan with a threshold for the variant allele fraction (VAF) of 10%. Variant calls in VCF format were then annotated through Annovar. As for Patients 1 and 2, we also performed NGS of pulmonary hypertension genes: ACVRL1, BMP10, BMPR2, CAV1, EIF2AK4, ENG, EPHB4, FOXF1, GDF2, GLMN, KCNK3, KRIT1, PTEN, RASA1, SMAD4, SMAD9, SOX17, TBX4, TEK (http://www.cgmc-psl.fr/spip.php?article45). This was added in the methods section of the manuscript lines 299-314.

NGS revealed a missense heterozygous variant in NKX2-1 c.440G>C p.(Gly147Ala). In the absence of another identified molecular

---

## [Decision Letter · Decision Letter 1]

23 Nov 2025

Deciphering the pathogenicity of three NKX2-1 variants in ultra-severe forms of childhood interstitial lung disease

PONE-D-25-14519R1

Dear Dr. SOREZE,

We’re pleased to inform you that your manuscript has been judged scientifically suitable for publication and will be formally accepted for publication once it meets all outstanding technical requirements.

Kind regards,

Andre van Wijnen

Academic Editor

PLOS ONE

Additional Editor Comments (optional):

Reviewers' comments:

Reviewer's Responses to Questions

**Comments to the Author**

Reviewer #1: All comments have been addressed

Reviewer #2: All comments have been addressed

2. Is the manuscript technically sound, and do the data support the conclusions?

Reviewer #1: Yes

Reviewer #2: Yes

3. Has the statistical analysis been performed appropriately and rigorously?

Reviewer #1: N/A

Reviewer #2: Yes

4. Have the authors made all data underlying the findings in their manuscript fully available?

Reviewer #1: Yes

Reviewer #2: Yes

5. Is the manuscript presented in an intelligible fashion and written in standard English?

Reviewer #1: Yes

Reviewer #2: Yes

Reviewer #1: The authors satisfyingly addressed my comments, and the revised manuscript has been substantially imoproved and is suitable for publication.

Reviewer #2: i have nothing else to say about this article, all my previous comments were addressed by the authors.

My main concerns were: the number of replicates, some poor descripted methods and a few points that were not fully backed by the results.

But with this revision all these points were covered, so I want to congratulate the authors on taking the time needed for this revision.

**Do you want your identity to be public for this peer review?** For information about this choice, including consent withdrawal, please see our Privacy Policy

Reviewer #1: **Yes: ** Olivier Danhaive, MD

Reviewer #2: No

---

## [Editor Report · Acceptance letter]

PONE-D-25-14519R1

PLOS One

Dear Dr. SOREZE,

I'm pleased to inform you that your manuscript has been deemed suitable for publication in PLOS One. Congratulations! Your manuscript is now being handed over to our production team.

Kind regards,

on behalf of

Dr. Andre van Wijnen

Academic Editor

PLOS One